# Textural Restoration of Broiler Breast Fillets with Spaghetti Meat Myopathy, Using Two Alginate Gels Systems

**DOI:** 10.3390/gels10010007

**Published:** 2023-12-21

**Authors:** Chaoyue Wang, Leonardo Susta, Shai Barbut

**Affiliations:** 1Department of Food Science, Ontario Agricultural College, University of Guelph, 50 Stone Road East, Guelph, ON N1G 2W1, Canada; chaoyue@uoguelph.ca; 2Department of Pathobiology, Ontario Veterinary College, University of Guelph, 50 Stone Road East, Guelph, ON N1G 2W1, Canada; lsusta@uoguelph.ca

**Keywords:** alginate gels, broiler meat, myopathy, spaghetti meat, texture improvement

## Abstract

The effects of salt-sensitive alginate (“A”) and a two-component salt-tolerant alginate system (“B”) used at a 0.5% or 1.0% level were evaluated in normal breast (NB) chicken fillets and in spaghetti meat (SM) fillets. Minced raw and cooked SM samples showed higher cooking loss (*p* < 0.05) and lower penetration force compared to NB meat. Both alginate systems significantly raised the penetration force in raw samples and decreased cooking loss (*p* < 0.05). Adding 1% of “A” or 0.5% “B” to SM, without salt, resulted in a similar penetration force as the cooked NB meat, while 1% “B” with salt resulted in a higher penetration force. Excluding salt from SM samples while adding alginate “A” or “B” improved texture profiles, but not to the same level as using NB without additives. Overall, salt, together with alginate “B”, improved the texture of SM to that of normal meat without myopathy.

## 1. Introduction

The annual global production of poultry meat is over 100 million tons; i.e., the amount has quintupled from half a century ago [1]. Increased demand for poultry meat required the industry to develop faster-growing broiler breeds. Today, broilers can be raised to a market weight of 4 kg (double that 50 years ago) in half the time to market [2,3]. Such rapid growth has also brought new challenges, such as the development of some myopathies that also affect meat quality. Breast meat is one of the most sought-after poultry products in the Western diet, and is also the most affected by myopathies. Wooden Breast (WB) and White Striping (WS) are two myopathies that have been known to affect several meat quality properties such as the texture of breast meat fillets, color, marinade uptake, water holding capacity, and cooking loss [4,5,6]. The quality deficiencies are currently quite substantial, and were estimated to cost USD >1 billion dollars/year to the North American poultry meat industry [7]. Currently, the industry is taking different approaches to overcome the meat quality issues, starting from breeding for less susceptible birds, feed adjustment to control growth rate, the addition of additives such as antioxidants, and further processing measures. At the meat processing level, the industry uses several technologies, such as blade tenderization, to mitigate some of the meat toughing (wooden breast) challenges [8]. 

While WB and WS are now well documented in terms of detection and mitigation [9,10], spaghetti meat (SM) is an emerging myopathy. WB and WS are sometimes associated with fast-growing, heavier birds, and result from muscle cell damage with insufficient time for repair. Oxidative stress and lack of oxygen supply to the fast-growing muscles can result in polyphasic myodegeneration, necrosis, and fibrosis with the accumulation of fat and connective tissue in the inner muscle tissue [11]. On the other hand, SM exhibits loose, detached fiber-like muscle fibers, mainly in the cranial part of the fillet. Histology studies have noted that SM featured hyalinization, poor fiber uniformity, and lower fat and connective tissue deposition [9]. Diminution in the density of connective tissue and higher amounts of inflammatory cell infiltrations resulted in muscle fiber detachment. NMR analysis revealed that SM fillets featured a higher proportion of extra-myofibrillar water in the superficial section, thus resulting in a reduction in the water-holding capacity. This makes SM different in appearance and texture compared to hard/tough WB and WS *Pectoralis major* fillets in which fibrosis is observed [12,13,14]. Spaghetti meat is found on the cranial surface layer of the fillet and is not mutually exclusive with the presence of other myopathies. It was revealed that SM could cause an increase in pH and lightness, decrease the water holding capacity, and therefore, lower the quality and functionality of the meat [6]. 

Processed poultry meat products include patties, nuggets, and sausages, where grinding/chopping can be used to mitigate some of the negative toughness/textural challenges associated with WB [15]. However, some textural differences may still be detected between normal fillets and fillets with myopathies, as Jarvis et al. [15] noted. Current mitigation techniques are mostly based on the breakdown of stiff muscle bundles by physical means to decrease the hardness of fillets with myopathies. However, as the characteristics of SM are largely the opposite of WB and WS, the mentioned mitigation techniques are not effective on fillets featuring this myopathy. Overall, the industry is looking for different methods to mitigate the product quality defects caused by SM, as corrective methods used for WS and WB are not appropriate.

Alginate, also known as alginic acid, is an anionic polysaccharide isolated and extracted from brown seaweeds (*Macrocystis pyrifera*, *Laminaria hyperborean*, or *Ascophyllum nodosum seaweeds)* under alkaline conditions. Alginate is a linear polymer formed by Guluronic (G) and Mannuronic (M) acids, where their ratio affects the gelling properties [16]. The two acids are arranged in an irregular pattern at different proportions of MM, MG and GG blocks. The mannuronic acid forms flexible linear structures with β-1,4 linkage. The guluronic acid forms an α-1,4 linkage, which provides a folded rigid structure and results in increasing the stiffness of the polymer chain. The important thing to note is that alginate forms a gel in the presence of divalent cations. The ions bind to the carboxyl groups in alginate and act as cross-linkers to form a gel network. The gelling process involves binding divalent ions across the GG block of aligned alginate chains. Therefore, the mannuronic acid vs. guluronic acid ratio affects the physicochemical properties of the resulting alginate gel. Higher M:G ratios form elastic gels, whereas low M:G ratios form dense and brittle alginate gels. The alginate gel formation process is often described by the “egg-box model” [17], which describes the cooperative mechanism of binding: two or more chains of guluronic acid units forming egg-box shapes, with interstices coordinated by calcium ions. The “Egg” describes the encapsulated molecules “packed” in this guluronic acid and calcium matrix. 

Alginate is often used in different foods as a thickening and/or gelling agent [18]. In products with only monovalent cations, alginates may not form gels and are used as a thickening agent [19]. Adding divalent cations such as Ca^2+^ could promote the speed of gel formation or allow alginate gel matrices to form in products otherwise lacking multivalent cations. One such application in the meat industry is the co-extrusion of sausages [20]. In this process, an alginate coating is sprayed on the extruded sausage. Sausages are then soaked in a water bath containing Ca^2+^ ions to promote alginate gelling, forming a solid film-like coating on the links. Another food product application using alginate and divalent cation supply is the creation of caviar-like spherical hydrogels with a liquid center [21]. Kumar et al. [22] reported that alginate could be added to low-fat ground pork patties to improve texture and water-holding capacity. The study concluded that alginate can be used as a fat replacement and help reduce cholesterol by 41~52%, while providing a similar texture to the original, full-fat product. One main advantage of alginate over other gums is its ability to form a gel without heating [19]. 

Restructured meats are defined as products made from multiple pieces of meat, using otherwise underutilized fabrication by-products [23]. The methods of making restructured meat include using the proteins from the smaller pieces to obtain cross-linking and/or using ingredients such as gums to create matrices to bind the system. According to the former, sodium chloride is used to extract the meat salt-soluble proteins (e.g., acting and myosin). Mechanical actions such as tumbling can be applied to speed up protein extraction. These proteins are then responsible for “gluing” the meat pieces during cooking [24]. Enzymessuch as transglutaminase could also be used to induce the binding of reformed meats. In that case, the enzyme induces a cross-bridging reaction between the glutamine and lysine, forming inter- and intramolecular glutamyl lysine bonds [25]. Gums such as alginate are already used in some restructured products. Overall, the three main components of sodium alginate, a calcium source and an acidifier are used to facilitate a slow calcium release [26]. Therefore, adding the appropriate amount of calcium can initiate cold gelation, bind the meat pieces and create the desired shape (when in a mold) and appearance, and later the product can be handled like a whole muscle. Lennon et al. [27] used alginate as a cold-set binding agent while reforming beef steaks. The experiment showed that alginate could be utilized to produce coherent steaks prior to cooking, and the steaks were also liked by consumers after cooking. Overall, this provided satisfactory texture results for reformed steaks that were comparable with those made with other cold-binding agents, such as transglutaminase and fibrinogen. 

Although alginate is used to improve the characteristics of meat products, there is a lack of documentation about its potential use to mitigate deficiencies in the functionality of the textural problems caused by SM myopathy. It should also be mentioned that studies showed that some primary meat processing procedures, such as stunning (i.e., caried out employing different currents and frequencies) can affect meat quality parameters [28]. Girasole et al. [29] concluded that a maximum rate of successful stun was obtained using a 400 Hz 200 mA current. Siqueira et al. [28] indicated that using 100 mA at 650 Hz promoted better meat quality. Furthermore, studies have revealed that the chilling method (air vs. water) also affects the percentage and severity of SM [30]. Therefore, our goal was to investigate ways to enhance/“repair” the texture of SM using two different alginate preparation systems. The alginate system “A” was a salt-sensitive one, while alginate “B” was not salt-sensitive. Samples were collected from the same processing line of a water-chilled processing plant on five different days. The parameters evaluated included the cold-binding of the raw minced normal fillets and fillets affected by SM, and later cooked samples prepared with or without salt. 

## 2. Result and Discussion

Adding alginate system “B” significantly (*p* < 0.05) increased the penetration force of raw minced NB and SM samples compared to the control (Figure 1), demonstrating the cold-binding effect of the gel. In NB samples without salt, 1% alginate “A” and 0.5% alginate “B” showed a similar penetration force. Adding alginate “B” to SM samples resulted in a greater penetration force compared to alginate “A”. The addition of salt to samples without alginate showed no significant difference from samples without salt. In contrast, the addition of salt to the alginate treatments lowered (*p* < 0.05) the penetration force compared to the unsalted counterparts with similar alginate treatments (Figure 1). In the case of porcine meat batters, similar results regarding the cold-binding ability of alginate and the lowered effectiveness of alginate in the presence of salt were observed [31]. Traditionally, alginate is known to be salt-intolerant because the presence of sodium ions competes with calcium ions as they have a similar charge. In any case, weak cross-links are formed when sodium ions bind to the alginate, resulting in a weaker gel structure or preventing gel matrix formation [32]. In our study, the two-component alginate gel system “B” overcame the salt tolerance of alginate by providing a higher amount of available cations.

Processes such as mincing, grinding, or blade tenderization mitigate WB fillets, characterized by the toughness of the fillet and deficits in the functionalities and textures of cooked meat, by shredding muscle bundles and diluting the surplus of collagens and fats into the batter, thus lowering the firmness of the batter. In our study, the penetration force for untreated SM cooked samples (3.76 N, Table 1) was significantly lower compared to the penetration force for NB samples (6.03 N). Similarly, Wang et al. [33] reported a lower compression force in cooked SM samples. Therefore, current processing methods alone could not be used to improve the textural deficiencies caused by SM, requiring an increase in the firmness of meat batters. Adding 1% alginate “A” and 0.5% alginate “B” to SM fillets resulted in similar penetration values compared to unsalted cooked NB samples. SM samples with 1% alginate “B” resulted in a higher penetration force than the NB samples without alginate. The penetration force for untreated cooked SM + salt samples (7.72 N, Table 1) was also significantly lower compared to the penetration force for NB + S samples (10.87 N). Adding salt improves the textural profile of meat products by extracting the salt-soluble protein and facilitating gel formation during cooking [34]. This effect is observed as the cooked samples with salt showed higher penetration force compared to the counterparts without salt. While improving the penetration force of samples, salt alone cannot be successfully used to mitigate the deficiencies of SM, as there was still a significant difference between NB and SM samples. Adding alginate “B” at 0.5% and 1% to SM samples with salt showed similar results compared to NB samples with salt. Also, treatment “B” at 1% resulted in a higher penetration force compared to 0.5%. As mentioned previously, treatment “A” does not tolerate salt and, therefore, was not evaluated with salt. Values between samples with and without salts were not directly compared in this study. Thus, alginate “A” at 1% or “B” at 0.5% performed the best in samples without salt, while treatment “B” at 1% showed a higher penetration force in samples with salt. 

Cooking losses differed (*p* < 0.05) between NB and SM samples without alginate (Table 1). Comparable trends were reported by Tasoniero et al. [14] for SM fillets. SM + salt samples resulted in significantly higher cooking loss compared to NB + salt. As indicated before, the cooking losses decreased when samples contained alginates, regardless of whether salt was present. Jeon et al. [35] found water-holding capacity to increase when alginate was added to meat batters used to make chicken patties. Yao et al. [36] investigated the effects of sodium alginate with different molecular weights on the water-holding capacity of myosin gels using chicken breast meat. The study noted that the major forces in alginate gel, electrostatic interactions and hydrogen bonding, were further enhanced with higher-molecular-weight (higher G:M ratio) alginate gel. This resulted in an increase in turbidity and a decrease in the surface hydrophobicity. The gel formed a heterogeneous network with larger cavities and, therefore, could hold more water molecules in the matrix. In our experiment, cooking losses were similar for NB and SM samples with alginate. The results demonstrate the ability of the alginate gel to mitigate the deficiencies of SM and improve the water holding capacity to comparable levels to NB.

Texture profile analysis of cooked samples was conducted to compare the NB and SM samples with or without alginate and salt. Hardness values were lower for SM samples (13.02 N without salt and alginate, Figure 2) than for NB samples (21.50 N). Hardness increased when alginate “B” was added to SM samples versus NB samples without alginate. Alginate “A” and “B” at 0.5% level did not affect the hardness of SM samples. Kumar et al. [22] and Jalal et al. [37] reported similar trends in texture improvements (i.e., hardness) with the addition of alginate to low-fat ground pork and beef patties. Salt addition increased hardness compared to counter samples without salt. SM + salt and alginate (“B” at 0.5%) were similar in hardness to NB samples with salt (no alginate). In another study on mitigating SM texture deficiencies, dairy proteins were added to meat batters containing 2% salt [33]. However, it should be noted that the dairy protein only formed gels upon heating (i.e., no binding in the raw state of the meat batter). Overall, the study reported that adding 2% whey protein isolate resulted in hardness and chewiness values similar to the cooked NB (33.93 vs. 34.98 N). In our study, adding 0.5% alginate “B” to SM resulted in 32.20 N vs. 30.29 N. Although both types of additives resulted in similar improvements in the texture of SM products, there are several advantages of using alginate, as it can provide binding (provide texture) to raw meat products with SM, as the cold-gelling effect was demonstrated in this experiment. Also, whey protein is recognized as an allergen and is, in fact, known to be the main allergen found in milk [38], and therefore alginate can be preferred as it did not show any allergic reaction [39]. Moreover, the cost of whey protein is estimated to be more than twice that of alginate, based on the literature [16,40]. Overall, adding 1% alginate “B” to SM + salt did not affect the hardness.

SM showed significantly lower springiness values compared to NB samples without salt. SM + 1% alginate “A” or 0.5% alginate “B” showed similar springiness compared to NB samples without alginate (Table 2). Lin and Mei [41] observed that alginate did not significantly affect the springiness of cooked reduced-fat pork meat batters. Dairy proteins added to modify the texture of poultry meat batters also showed limited effects on the springiness of the cooked minced meat [42]. In the present study, there was no significant difference in springiness between SM + salt and NB + salt (Table 3). Although the results from the studies show that alginate had a limited effect on NB, it is worth noting that, in our study, adding alginate improved the SM to yield a similar performance as NB samples. This shows that alginate gel could indeed be used to mitigate textural deficiencies caused by SM.

In all cases, SM samples showed significantly lower cohesiveness and chewiness values compared to NB samples, regardless of any alginate addition. When salt was added, SM +/− alginate revealed lower cohesiveness compared to NB samples without alginate. SM + 0.5% alginate “B” showed similar chewiness as the NB samples. Similar to other reports, adding alginate had a limited effect on cohesiveness, chewiness, and resilience, as observed in different studies using alginate to bind meat patties [36] or using dairy proteins in chicken patties. However, it is important to note that in our study, adding alginates improved the otherwise deficient texture parameters (e.g., penetration force and hardness) of SM samples to comparable levels of normal minced fillets. In contrast, other studies focused on the effects of the non-meat additives without deficiencies caused by SM.

## 3. Conclusions

Spaghetti meat (SM) samples with or without salt showed lower raw and cooked penetration force, hardness, cohesiveness, and chewiness values, as well as greater cooking loss compared to NB samples. Alginate and/or salt addition increased the penetration force for both raw and cooked samples. Penetration force was similar in both alginate systems (“A” and “B”) added to SM and NB samples. Alginate addition to all samples almost eliminated cooking loss. Without adding salt, 1% alginate “B” treatment showed the best improvement when added to SM, while the 0.5% alginate “B” treatment showed the most significant improvement in the SM + salt meat batter. Salt is also an important factor in mitigating textural deficiencies in cooked SM samples; adding salt resulted in similar texture profiles to NB samples with salt. Overall, alginate was demonstrated to be able to mitigate the texture and functional deficiencies of raw and cooked SM samples manufactured with and without salt. Further experiments are planned to validate the performance of alginate-mitigated SM products subjected to different processing procedures. It may also be pertinent to examine the effects of emerging technologies such as high-hydrostatic-pressure processing (e.g., known to induce some protein gelation, as well as microbial inactivation) on alginate–SM meat systems.

## 4. Material and Methods

### 4.1. Meat Sample Preparation

Normal breast (NB) fillets and fillets showing only the SM myopathy were collected from five different flocks at a commercial water chill poultry plant. Twenty breast fillets were used per flock for each treatment in each of the five independent trials. Because SM is often only found at the top half of the cranial part of the fillet, only the upper parts of SM and NB fillets were used. These sections were minced for 15 s using a food processor (Braun, model UK1-Type 4259, Kronberg, Germany). Two alginate systems were used (“A” = Eurogum A/S, Eurogel MBA 3006, Herlev, Denmark, and “B” = Eurogel MBA 5081 plus Eurogel MBA 5082). Additionally, to prepare alginate “B”, we tested the effect of adding salt (NaCl) to the alginate preparation. This was only done when preparing alginate “B” samples because this is a salt-tolerant alginate, whereas “A” is not (Table 1). Samples were identified with letters indicating the type of breast meat used (NB or SM), type of alginate treatment, and if salt (1.5%) was added (designated as +S). Treatment “A” (a single-component alginate system) was added at 1%, and treatment “B” (a two-component system) was added at 0.5% or 1%. Meat (133.5 g), alginate, and water were mixed to achieve a total weight of 150 g. Samples were mixed manually at room temperature for 30 s using a spatula. The batters were packed in four 50 mL plastic test tubes containing 30 g each and stored for 24 h at 4 °C to allow alginate to cold-gel. 

### 4.2. Texture Analysis

A penetration test was then performed using a texture analyzer (Stable Micro System TA.XT2, Texture Technologies Corp., Scarsdale, NY, USA) equipped with a 10 mm cylindrical flat stainless steel probe, at a speed of 1.5 mm/s, to a distance of 15 mm in duplicates. All samples were then cooked in a water bath to an internal temperature of 72 C and then stored for 24 h at 4 °C, before cooking loss was determined (*n* = 4/treatment), followed by a penetration test (*n* = 2/treatment). The cooking losses of the samples were calculated by subtracting the weights of raw samples from the cooked ones. Cooked samples were also cut into pucks (20 mm diameter and 10 mm height) to undertake the texture profile analysis by applying a two-cycle 50% compression cylindrical large plate at a crosshead speed of 1.5 mm/s.

### 4.3. Statistical Analysis

The experiment was designed as a complete randomized block with five replications, where flocks were considered the random effect and formulations the fixed effect. Data were analyzed using the one-way ANOVA option of the GLM procedure present in the SAS software 9.4 (SAS Institute Inc., Cary, NC, USA). Tukey’s HSD test for multiple comparisons was used to compare the sample means. Samples with salt and samples without salt were analyzed separately because the two groups represent different categories of products, and were not meant to be compared side by side. Samples were considered to be significantly different when *p* < 0.05. Data are reported as means ± SEM.

## Figures and Tables

**Figure 1 gels-10-00007-f001:**
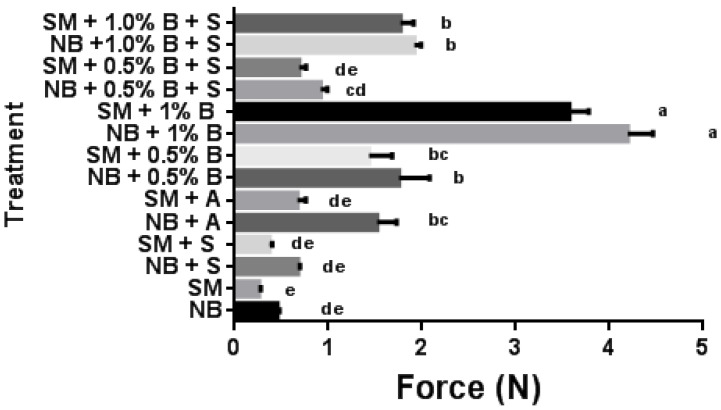
Penetration force values of raw normal breast (NB) and spaghetti meat (SM) minced samples with or without alginates (“A” and “B”), and with (1.5%) or without salt (S). ^a–e^ Means (*n* = 10) followed by a different letter superscript are significantly different (*p* < 0.05). Bars show standard error.

**Figure 2 gels-10-00007-f002:**
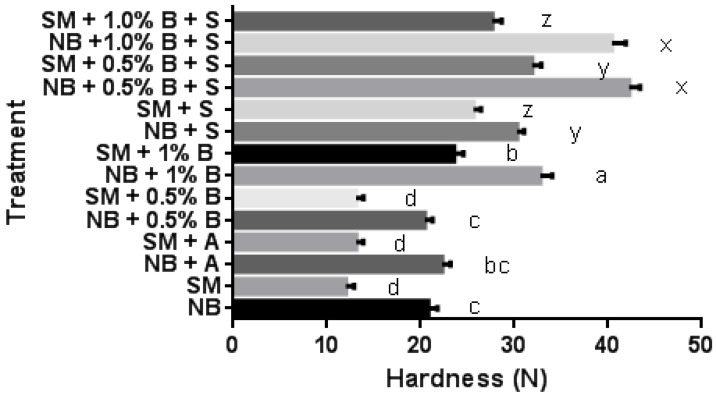
Hardness of values for cooked normal breast (NB) and spaghetti meat (SM) minced samples with and without alginate treatment (“A” and “B”), and with (1.5%) or without salt (S). Samples without salt (separated by letters a–d) and with salt (separated by letters x–z) were not analyzed together. Means (*n* = 30) followed by a different letter are significantly different (*p* < 0.05). Bars show standard error.

**Table 1 gels-10-00007-t001:** Effects of chicken breast myopathy (NB = normal breast; SM = spaghetti meat) and alginate system (noted as “A” and “B”) on penetration force and cooking loss of the minced meat.

	Sample Type							
Treatment	NB	SM	NB + 1% “A”	SM + 1% “A”	NB + 0.5% “B”	SM + 0.5% “B”	NB + 1% “B”	SM + 1% “B”
NO SALT ADDED								
Penetration Cooked (N)	6.03 ± 0.30 ^a^	3.76 ± 0.22 ^b^	7.58 ± 0.28 ^c^	4.85 ± 0.31 ^ab^	8.08 ± 0.38 ^c^	5.59 ± 0.36 ^a^	10.03 ± 0.50 ^d^	8.04 ± 0.36 ^c^
Cooking loss (%)	6.72 ± 0.39 ^a^	10.03 ± 0.57 ^b^	0.60 ± 0.07 ^cd^	0.90 ± 0.12 ^cd^	0.69 ± 0.11 ^cd^	1.37 ± 0.22 ^c^	0.11 ± 0.03 ^d^	0.08 ± 0.02 ^d^
SALT ADDED								
Penetration Cooked (N)	10.87 ± 0.45 ^a^	7.72 ± 0.54 ^b^	N/A	N/A	12.95 ± 0.57 ^c^	10.02 ± 0.50 ^a^	13.64 ± 0.45 ^c^	10.55 ± 0.46 ^a^
Cooking loss (%)	1.20 ± 0.18 ^a^	2.37 ± 0.23 ^b^	N/A	N/A	0.06 ± 0.01 ^c^	0.13 ± 0.03 ^c^	0.10 ± 0.02 ^c^	0.09 ± 0.02 ^c^

^a–d^ Means of penetration force ± standard error (*n* = 10), and cooking loss (*n* = 20) followed by a different superscript in a given row, are significantly different (*p* < 0.05). N/A = Not applicable, as alginate “A” does not tolerate salt.

**Table 2 gels-10-00007-t002:** Effect of the cooked chicken breast myopathies (NB = normal breast; SM = spaghetti meat) and alginate system (noted as “A” and “B”) on the texture of minced meat without salt.

	Sample Type							
Treatment	NB	SM	NB + 1% “A”	SM + 1% “A”	NB + 0.5% “B”	SM + 0.5% “B”	NB + 1% “B”	SM + 1% “B”
Springiness	0.74 ± 0.01 ^b^	0.72 ± 0.01 ^c^	0.78 ± 0.01 ^a^	0.76 ± 0.01 ^ab^	0.76 ± 0.01 ^ab^	0.75 ± 0.01 ^abc^	0.78 ± 0.01 ^ab^	0.78 ± 0.01 ^a^
Cohesiveness	0.57 ± 0.01 ^a^	0.51 ± 0.01 ^b^	0.45 ± 0.01 ^c^	0.39 ± 0.01 ^d^	0.43 ± 0.01 ^c^	0.38 ± 0.01 ^de^	0.40 ± 0.01 ^d^	0.35 ± 0.01 ^e^
Chewiness	9.04 ± 0.48 ^b^	4.77 ± 0.43 ^d^	7.88 ± 0.43 ^bc^	4.26 ± 0.38 ^d^	6.81 ± 0.26 ^c^	3.99 ± 0.25 ^d^	10.41 ± 0.41 ^a^	6.89 ± 0.79 ^c^
Resilience	0.18 ± 0.01 ^a^	0.16 ± 0.01 ^b^	0.15 ± 0.01 ^b^	0.12 ± 0.01 ^cd^	0.15 ± 0.01 ^bc^	0.12 ± 0.01 ^d^	0.15 ± 0.01 ^b^	0.14 ± 0.01 ^bcd^

^a–e^ Means ± standard error (*n* = 18) followed by a different superscript in a given row are significantly different (*p* < 0.05).

**Table 3 gels-10-00007-t003:** Effect of chicken breast myopathies (NB = normal breast, SM = spaghetti meat) and alginate system (noted as A and B) on texture of minced meat with salt.

	Sample Type					
Treatment	NB + Salt	SM + Ssalt	NB + 0.5% “B” + Salt	SM + 0.5% “B” + Salt	NB + 1.0% “B” + Salt	SM + 1.0% “B” + Salt
Springiness	0.83 ± 0.01 ^a^	0.83 ± 0.01 ^a^	0.82 ± 0.01 ^a^	0.84 ± 0.01 ^a^	0.82 ± 0.01 ^a^	0.79 ± 0.01 ^b^
Cohesiveness	0.53 ± 0.01 ^a^	0.47 ± 0.01 ^b^	0.51 ± 0.01 ^a^	0.47 ± 0.01 ^b^	0.49 ± 0.01 ^ab^	0.40 ± 0.01 ^c^
Chewiness	13.25 ± 0.43 ^b^	10.11 ± 0.41 ^c^	18.41 ± 0.79 ^a^	12.80 ± 0.56 ^b^	17.73 ± 0.96 ^a^	8.92 ± 0.52 ^c^
Resilience	0.19 ± 0.01 ^ab^	0.17 ± 0.01 ^bc^	0.20 ± 0.01 ^a^	0.19 ± 0.01 ^ab^	0.20 ± 0.01 ^a^	0.15 ± 0.01 ^c^

^a–c^ Means ± standard error (*n* = 18) followed by different superscript in a given row are significantly different (*p* < 0.05).

## Data Availability

The data presented in this study are available on request from the corresponding author. The data are not publicly available due to confidentiality agreement with the processing plant where samples were obtained.

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
