# Peer review of "Textural Restoration of Broiler Breast Fillets with Spaghetti Meat Myopathy, Using Two Alginate Gels Systems"

_gels, 2023, doi:10.3390/gels10010007_

Round 1

Reviewer 1 Report

Comments and Suggestions for Authors

1. In line of 89, the expression of Ca++ is suggested to be revised as Ca2+.

2. The background of alginate ‘A’ and ‘B’ is suggested to be added.

3. The basic principle and the design of the work are suggested to be

introduced.

Author Response

Dear reviewer,

Thank you for your suggestions. To your comments, the following edits were made:

Comments and Suggestions for Authors

  1. In line of 89, the expression of Ca++ is suggested to be revised as Ca2+.

Agreed, modification made

  1. The background of alginate ‘A’ and ‘B’ is suggested to be added.

Added information about alginate ‘a’ and ‘b’ in the introduction.

  1. The basic principle and the design of the work are suggested to be

introduced.

Added an explanation of the experimental design sample collection to the introduction section.

Thank you.

Reviewer 2 Report

Comments and Suggestions for Authors

The authors describes the results of investigate ways to improve the texture of Spaghetti Meat (SM) fillets way two different alginate preparation systems, salt sensitive and not salt sensitive, as salt is used in processed meats. The results are interesting because deal with an important topic for poultry food industry. I think that the manuscript can be improved with some items indicated below.

This paper could be cited to improve the introduction regard poultry quality woth slaughter aspects discussion. I think it is interesting to discuss data with paper below:

Effect of electrical water bath stunning on physical reflexes of broilers: evaluation of stunning efficacy under field conditions

M Girasole, R Marrone, A Anastasio, A Chianese, R Mercogliano, ...

Poultry science 95 (5), 1205-1210

The authors also must indicate how were fillets sampled . Which experimental design? directly or take from local market?

Comments on the Quality of English Language

.

Author Response

Dear reviewer,

Thank you for your suggestions. The following edits were made based on the suggestions.

Comments and Suggestions for Authors

The authors describes the results of investigate ways to improve the texture of Spaghetti Meat (SM) fillets way two different alginate preparation systems, salt sensitive and not salt sensitive, as salt is used in processed meats. The results are interesting because deal with an important topic for poultry food industry. I think that the manuscript can be improved with some items indicated below.

This paper could be cited to improve the introduction regard poultry quality woth slaughter aspects discussion. I think it is interesting to discuss data with paper below:

Effect of electrical water bath stunning on physical reflexes of broilers: evaluation of stunning efficacy under field conditions

M Girasole, R Marrone, A Anastasio, A Chianese, R Mercogliano, ...

Poultry science 95 (5), 1205-1210

Added a discussion, including the above paper, regarding the effect of stunning on the quality of chicken breast meat to the Introduction.

The authors also must indicate how were fillets sampled . Which experimental design? directly or take from local market?

Added a paragraph explaining the sampling method and experiment design to the Introduction and to the Materials and Methods.

Reviewer 3 Report

Comments and Suggestions for Authors

Introduction

Shorten text about alginates

Insert more text (recent publications) about meat restructuring methods

Results

Need more research data on the effect of gel on organoleptic properties, pH, color, etc.

How does the gel affect water binding capacity?

Conclusion

Adjust the conclusion. Remove unnecessary general text. Describe future research in this area.

Author Response

Dear reviewer,

Thank you for your suggestion. The following edits were made based on the suggestions.

Comments and Suggestions for Authors

Introduction

Shorten text about alginates

Insert more text (recent publications) about meat restructuring methods

Results

The paragraph on application of alginate in different food product was shortened from the Introduction.

Extended the paragraph on methods to form restructured meat.

Need more research data on the effect of gel on organoleptic properties, pH, color, etc.

How does the gel affect water binding capacity?

In this study, we have not specifically examine the organoleptic properties, as alginate is already being used by the meat industry in various products. We hope to have more funding in the future to also look at this aspect.

Conclusion

Adjust the conclusion. Remove unnecessary general text. Describe future research in this area.

Removed text on comparison between advantages of alginate vs. dairy proteins from the Conclusion. Added suggestions for pertinent future researches in the area.

Reviewer 4 Report

Comments and Suggestions for Authors

The manuscript is interesting but falls short in its analysis. It has the potential to enhance the depth of its discussion and analysis, making a more comprehensive and informative contribution to the field of broiler breast fillet texture improvement in the context of Spaghetti Meat Myopathy. It would be great if the author included some photographs and additional functional properties such as color, pH, and protein solubility to further enhance and support the texture data.

Method for cooking losses were not properly described in the methodology.

Author Response

Dear reviewer,

Thank you for your suggestions. The following edits were made based on the suggestions.

Comments and Suggestions for Authors

The manuscript is interesting but falls short in its analysis. It has the potential to enhance the depth of its discussion and analysis, making a more comprehensive and informative contribution to the field of broiler breast fillet texture improvement in the context of Spaghetti Meat Myopathy. It would be great if the author included some photographs and additional functional properties such as color, pH, and protein solubility to further enhance and support the texture data.

Photos of chicken breast fillet with SM myopathy and the samples made from normal vs SM fillets are included in the graphical abstract. We are open to see if the editor would like to include them in  the main text

Added several lines, explaining the effect of SM on pH, color, and water holding capacity (Based on one of our previous studies, focusing on describing the physico-chemical properties of myopathies on chicken breast fillet)

Method for cooking losses were not properly described in the methodology.

A detailed description of calculating the cooking loss has been added to the methodology.

Round 2

Reviewer 1 Report

Comments and Suggestions for Authors

I think the manuscript could be  accepted in present form.

Reviewer 3 Report

Comments and Suggestions for Authors

All corrections are inserted